# How Climate Change Impacts the Distribution of *Lithocarpus hancei* (Fagaceae), a Dominant Tree in East Asian Montane Cloud Forests

Yongjingwen Yang [1,2,3,4,5], Lin Lin [1,2,3], Yunhong Tan [4,5,*] and Min Deng [1,2,3,*]

1   School of Ecology and Environmental Sciences, Yunnan University, Kunming 650500, China; yangyongjingwen@gmail.com (Y.Y.); linlin.ynu@gmail.com (L.L.)
2   Yunnan Key Laboratory of Plant Reproductive Adaptation and Evolutionary Ecology and Institute of Biodiversity, Yunnan University, Kunming 650500, China
3   The Key Laboratory of Rare and Endangered Forest Plants of National Forestry and Grassland Administration & The Key Laboratory for Silviculture and Forest Resources Development of Yunnan Province, Yunnan Academy of Forestry and Grassland, Kunming 650201, China
4   Southeast Asia Biodiversity Research Institute, Chinese Academy of Sciences & Center for Integrative Conservation, Xishuangbanna Tropical Botanical Garden, Chinese Academy of Sciences, Menglun 666303, China
5   Yunnan International Joint Laboratory of Southeast Asia Biodiversity Conservation, Menglun 666303, China
*   Correspondence: tyh@xtbg.org.cn (Y.T.); dengmin@ynu.edu.cn (M.D.); Tel.: +86-135-7814-7508 (Y.T.); +86-136-6172-0514 (M.D.)

**Abstract:** Subtropical montane moist evergreen broadleaf forests (MMEBFs) have a unique environmental envelope harboring a high percentage of endemic biota. This ecosystem is highly vulnerable, and little is known about its possible response to future climate change. In this study, we used *Lithocarpus hancei* (Fagaceae), a dominant tree in East Asian subtropical MMEBFs, as a proxy to investigate MMEBF distribution dynamics and key distribution restriction factors. In total, 607 *L. hancei* occurrence points were obtained after being gathered and filtered. The random forest model was selected as the best model. Eight environmental variables were selected to simulate the potential suitable areas of *L. hancei* under the Last Glacial Maximum (LGM), present, and future (2041–2060, 2061–2080, 2081–2100) climate change scenarios, including four shared socioeconomic pathways (SSP1-2.6, SSP2-4.5, SSP3-7.0, SSP5-8.5). Our results showed that precipitation of the warmest quarter, the precipitation of the driest month, the mean diurnal range, and temperature seasonality are the key factors affecting the spatial range of *L. hancei* with 11.2%, 10.9%, 8.1%, and 7.6% contributions, respectively. The current distribution of *L. hancei* is mainly within East and South China, with a scattered distribution in North Indo-China and the Southeast Himalayas. The highly suitable area only accounts for 9.7% of the total distribution area. The distribution area of the current suitability area is the smallest compared to that under LGM and future scenarios. In all the future climatic scenarios, the highly suitable areas of *L. hancei* would decrease or even disappear, whereas the medium- and low-suitability areas might increase with the centroid of the total suitable area northern. Its distribution in Central China, the Southern Himalayas, and Northern Indo-China will be lost in the future. Overall, our study predicted a prominent degradation of East Asian MMEBFs in the future. In situ and ex situ conservation on East Asian MMEBFs should be prioritized and enforced.

**Keywords:** species distribution modeling; bioclimatic variables; (sub) tropical montane cloudy forests; East Asia subtropics

## 1. Introduction

Evergreen broadleaved forests (EBFs) are major ecosystems in the East Asian subtropics [1] and exhibit high species diversity, productivity, and richness, which contributes to both the biodiversity and sustainable development of the subtropical regions of China [2].

Montane moist evergreen broadleaf forests (MMEBFs), also known as montane cloud forests, are a subtype of EBFs mainly distributed in Southern China [3], at altitudes of about 1800–3400 m [4]. Compared with other EBFs, MMEBFs have a unique microclimate envelope characterized by high humidity throughout the year, relatively cool temperatures, and a windy environment. Future global warming and increased aridity might increase the incidence and strength of extreme climate events, e.g., wildfires, floods, and drought, which can severely impact the distribution of MMEBFs, leading to the prominent degradation of forests. On the other hand, Asian EBFs are facing accelerating human disturbance because of their high productivity and species richness. The low-land areas are mostly taken away by agriculture. The nature reserves are mainly distributed along the middle to upper portions of mountains resulting in a highly fragmented habitat of MMEBFs [3]. The endemic biota of MMEBFs will face more severely endangered conditions. Nevertheless, how global climate change impacts this unique forestry ecosystem remains largely unknown. Understanding the possible distribution dynamics of the dominant canopy trees in the MMEBFs ecosystem is essential to establishing guidance for forestry conservation management to safeguard the long-term survival and health of this unique ecosystem, especially during the Anthropocene area.

*Lithocarpus hancei* (Fagaceae) is an evergreen tree widespread in the East Asian subtropics from the East Himalayas and Northern Indo-China to Southern China [5] (Figure 1). It is shade-tolerant and commonly found in the forest shade layer or canopy gaps [6], with a wide distribution within an altitudinal range of approximately 800–2600 m. Its core distribution area is located in South China at an intermediate altitude, with an upper elevation limit typically below 1500 m [7], but it can stretch to a higher elevation range of 2000–2900 m in the East Himalayas and the Northern Indo-China Peninsula. *L. hancei* is among the dominant regional canopy trees of MMEBFs [8]. Its habitats are characterized by high humidity, cool temperatures, and strong winds, especially in the upper mountain areas [9]. The unique microclimates of MMEBFs usually form a very specialized landscape—montane moss dwarf forests [10]. The biota restrictively distributed in MMEBFs typically have very special ecological requirements and occupy delicate niches. Therefore, MMEBFs harbor high proportions of endemic and endangered species. All these factors lead to MMEBFs being among the most vulnerable forestry ecosystems worldwide [11].

Species distribution models (SDMs) were initially developed to study changes in species distributions and diversity patterns under climate change and have been applied to study the relationship between species distributions and environmental variables [12]. They are powerful tools widely applied in ecology [13], evolutionary biology [14], and global change biology [15]. In recent years, multiple SDMs have been implemented in R (https://www.r-project.org/, accessed on 16 August 2022) to conveniently analyze the current distribution of species and environmental factors. These models can link current species distribution data to environmental variables [16] and thus estimate the niches of species according to specific algorithms. Then, projections can be made to explain the probability of species emergence and habitat suitability throughout the landscape [17], which can significantly improve the efficiency of forestry management and conservation.

In this study, we used *L. hancei* as a proxy to infer the potential distribution of MMEBFs under different climate change scenarios during the Last Glacial Maximum (LGM), present, and three future time periods (2041–2060, 2061–2080, 2081–2100), aiming to accomplish the following: (1) reveal the key restrictive climatic factors impacting the distribution of *L. hancei*; (2) characterize its distribution dynamics and demographic change from the LGM to the present and then to the future under different $CO_2$ emission scenarios. This study provides crucial information on the possible response of dominant canopy trees of East Asian MMEBFs to better conserve this unique ecosystem.

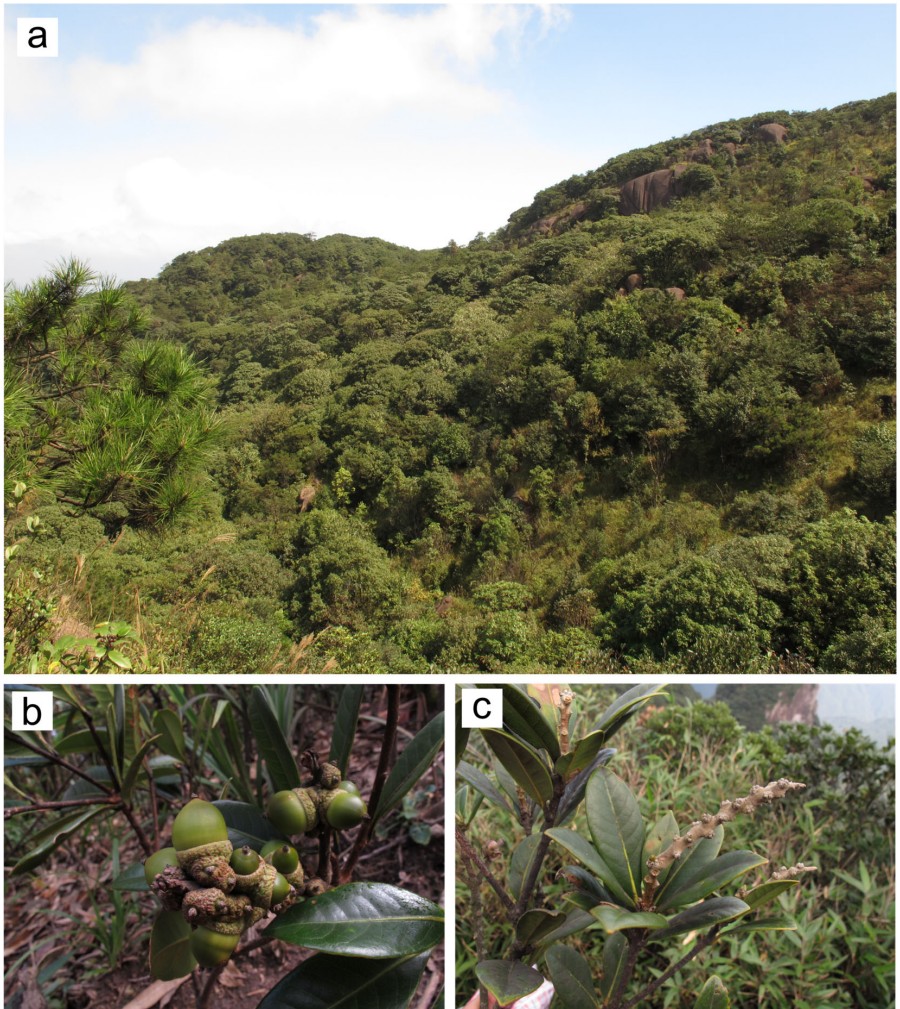

**Figure 1.** (**a**) Landscape of middle-elevation montane where could forests dominate by *Lithocarpus hancei* (Meihua Mt. National Nature Reserve, Fujiang province, China); (**b**) Twigs of *L. hancei*; (**c**) Inflorescence of *L. hancei* (Mangshan, National nature reserve, Hunan province).

## 2. Materials and Methods

### 2.1. Research Area

The current core distribution of *L. hancei* is located in Southeast China and East China, extending into the low-altitude valleys of the Southeast Himalayas and Northern Indo-China (Figure 2).

### 2.2. Data Collection

#### 2.2.1. Occurrence Data and Data Filtering Process

The *L. hancei* occurrence data were obtained from the Chinese Virtual Herbarium (CVH; https://www.cvh.ac.cn/, accessed on 16 August 2022) and the Global Biodiversity Information Facility (GBIF; https://www.gbif.org/, accessed on 16 August 2022). We only kept occurrence records with coordinates plus images. Unreliable data points were removed if they exhibited incorrect georeferencing (e.g., coordinates falling in the sea or outside the known distribution range) or mistaken taxonomic identification.

We manually checked the detailed collection sites for the herbarium sheets without GPS coordinates in the collection records to locate the specific sampling sites. Then, we determined the corresponding GPS coordinates on Google Earth v2017. Excessive inhomogeneity of occurrence data can lead to bias in simulation results [18]. Thus, the 'dismo' package [19] in R v4.2.2 [20] was used to filter the points with excessively dense

occurrence data so that only one record was randomly retained in each 0.2° × 0.2° grid. In total, 607 occurrence points of *L. hancei* were obtained after thinning (Table S1).

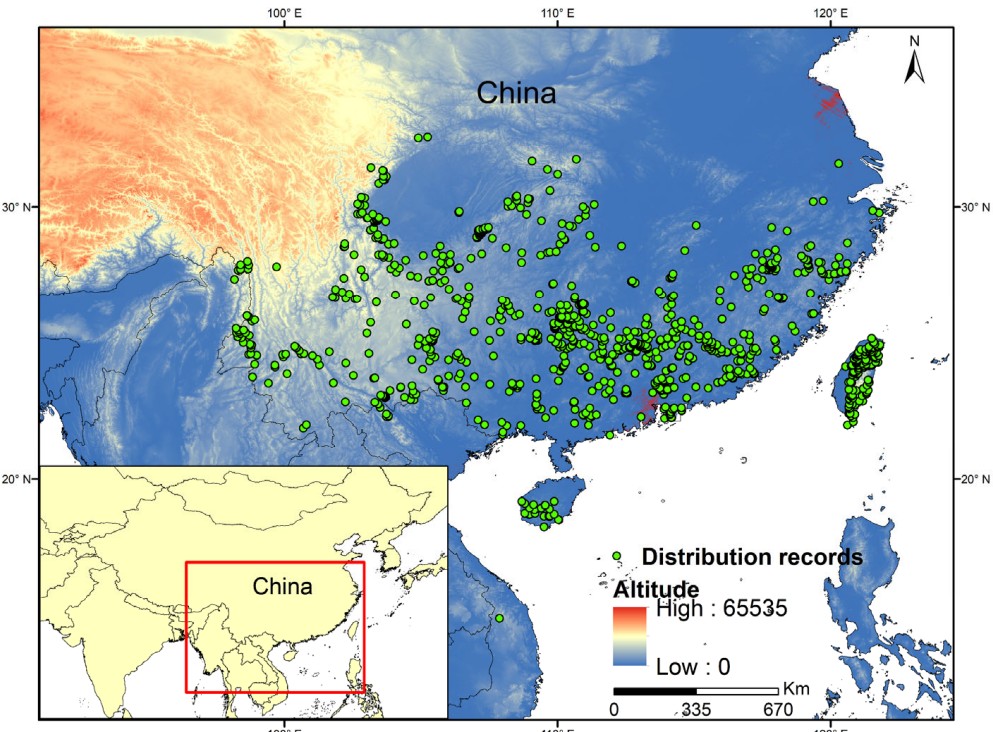

**Figure 2.** The current distribution records of *Lithocarpus hancei*.

### 2.2.2. Environmental Variables

Data for 39 environmental variables were downloaded, including 19 climatic and 20 soil variables. We used the World Climate Database version 2.1 (http://www.worldclim.org/, accessed on 16 August 2022) [21] to access the bioclimatic variables for current and future projections. The multi-model median ensemble of the Global Climate Models (GCMs) was obtained for the 19 climatic variables based on the median of three GCMs for the LGM provided by the Paleoclimate Modeling Inter-comparisons Project Phase 5 [22], i.e., CCSM4, MIROC-ESM, and MPI-ESM-P, respectively [23]. For future climate projections, data was obtained from a database containing bioclimatic variables for nine GCMs of the CMIP6 [24] for four Shared Socioeconomic Pathways (SSPs). The 19 current climate bioclimatic variables were downloaded at a 2.5 arc-minute resolution (approximately 5 km at the equator). For mapping the suitable climatic envelope of *L. hancei* under the impact of climate change, we used data from three GCMs, i.e., BCC-CSM2-MR, CNRM-CM6-1, and CanESM5, for three future periods (each at least 20 years from the present): 2041–2060, 2061–2080 and 2081–2100. Within each future GCM, there are four SSPs, ranging from SSP1-2.6 (aggressive mitigation/lowest emissions) to SSP5-8.5 (highest emissions scenario). In order to reduce the uncertainty associated with a particular GCM in the three GCMs, we averaged the occurrence probability. Soil variables were obtained from the Harmonized World Soil Database (HWSDV.1.2; http://www.fao.org/, accessed on 16 August 2022) with a spatial resolution of 30 arc-second.

To standardize the resolution of environmental variables, we imported the raster layers of the 39 environmental variables into ArcGIS (ESRI Inc., Redlands, CA, USA) for reclassification, and the unified resolution was 2.5 arc-minute.

### 2.3. Variable Selection

Although 39 environmental variables were available, most of them are highly correlated within the study area, which can lead to model overfitting. To avoid such overfitting owing to

the high collinearity of environmental variables [25], we performed environmental variable contribution and correlation analysis. All environmental variables were first added to the random forest model and replicated 10 times. The contribution of each bioclimatic variable was obtained using the "getVarImp" function in the "sdm" package in R (Table S2). As the contributions of soil and terrain variables were minor (ranging from 0.3 to 0), we kept only 19 climatic variables for the subsequent analyses. Pearson's rank correlation analysis was then conducted for different climate variables using the "findCorrelation" function in the "caret" package [26] in R. The important climatic variables were then selected according to their relative contributions (Table S2) and the results of Pearson's rank correlation analyses (Table S3). Combined with the contribution of environmental factors, we retained factors with correlation coefficients under 0.8. A couple of environmental factors had correlation coefficient values greater than |0.8|, after which only one variable with a higher contribution was retained and used in the SDMs [27,28]. Finally, AMT (annual mean temperature, °C), MDR (mean diurnal range, °C), TS (temperature seasonality, %), MTWM (max temperature of warmest month, °C), PrecWM (precipitation of wettest month, mm), PrecDM (precipitation of driest month, mm), PS (precipitation seasonality, mm), and PrecWQ (precipitation of warmest quarter, mm) were selected for the subsequent distribution simulation.

### 2.4. Model Evaluation and Modeling Simulation

The 'sdm' [29] package in R was used to select the best model for ecological niche modeling and simulate the potential distribution area for prediction analysis. This package enables the application of 21 models: random forest (RF), maximum entropy (MaxEnt), boosted regression tree (BRT), classification and regression trees (CART), flexible discriminant analysis (FDA), generalized additive model (GAM), generalized linear model (GLM), multivariate adaptive regression spline (MARS), mixture discriminant analysis (MDA), support vector machine (SVM), multi-layer perceptron (MLP), radial basis function (RBF), recursive partitioning and regression trees (Rpart), Bioclim, Bioclim.dismo, Domain.dismo, Glmnet, Mahalanobis, Ranger, Maxlike. The performance of each model was evaluated for *L. hancei*. All models used 70% of the distribution data as the training set and the remaining 30% as the test set, with 10 replicates and cross-validation as run type. The training dataset was used for model calibration, and the testing dataset was used for cross-validation of the model evaluation during the calibration process. The accuracy of the models was evaluated using the area under the receiver operating characteristic curve (AUC) and the true skill statistic (TSS) [30]. Usually, when the AUC value is between 0.8 and 0.9, the prediction result is considered accurate. When the AUC value is in the range of 0.9 to 1.0, the prediction result is considered particularly good. TSS is a threshold-dependent metric, where a TSS value > 0.6 is considered excellent [31]. Thus, higher AUC and TSS values indicate higher accuracy. RF was the model with the highest AUC and TSS values and was therefore selected for the subsequent SDM analysis.

### 2.5. Analysis of Habitat Change and Range Contraction or Expansion

After using RF to model the species distribution, the future spatial changes compared to the current distribution were calculated by ArcGIS 10.8 (ESRI Inc.). The current and future range results were imported into ArcGIS in ASCII format and reclassified into four regions according to the predicted existence probability 0 to 1 using the natural breakpoint method as follows: unsuitable habitats (0–0.1), low-suitability habitats (0.1–0.3), medium-suitability habitats (0.3–0.6), and high-suitability habitats (0.6–1) [32]. Then, we used SDMtoolbox 2.0 (http://www.sdmtoolbox.org/, accessed on 16 August 2022) [33] in ArcGIS to perform area expansion (contraction) analysis to estimate the suitable areas. The forecast map was converted into a binary raster layer (inappropriate, 0; suitable, 1). By comparing the potential distributions of suitable/unsuitable layer values under different climate scenarios, a total of four distribution changes were identified, namely, contraction, expansion, no change, and no occupation. In addition, the range of our prediction was

based on the latitudes and longitudes of the collected records, and a circle of 5° was set as a buffer around each distribution point as the area used for calculation. Additionally, the west distribution range extends to the vicinity of the Bay of Bengal.

### 2.6. Centroid Change

Area calculations and centroids for different suitability classes were calculated using SDMtoolbox in ArcGIS. In addition, the changes in the area and centroid position of different categories of suitable areas under the different climate change scenarios were compared.

## 3. Results

### 3.1. Model Performance

Evaluation of the 21 models showed that RF had the highest AUC and TSS values (AUC = 0.96, TSS = 0.82) (Table S4). Therefore, it was selected for subsequent predictions of *L. hancei* distributions. Cross-validation of the known occurrence points of *L. hancei* to its predicted suitable habitat during the current period based on RF were well matched (Figure 3a).

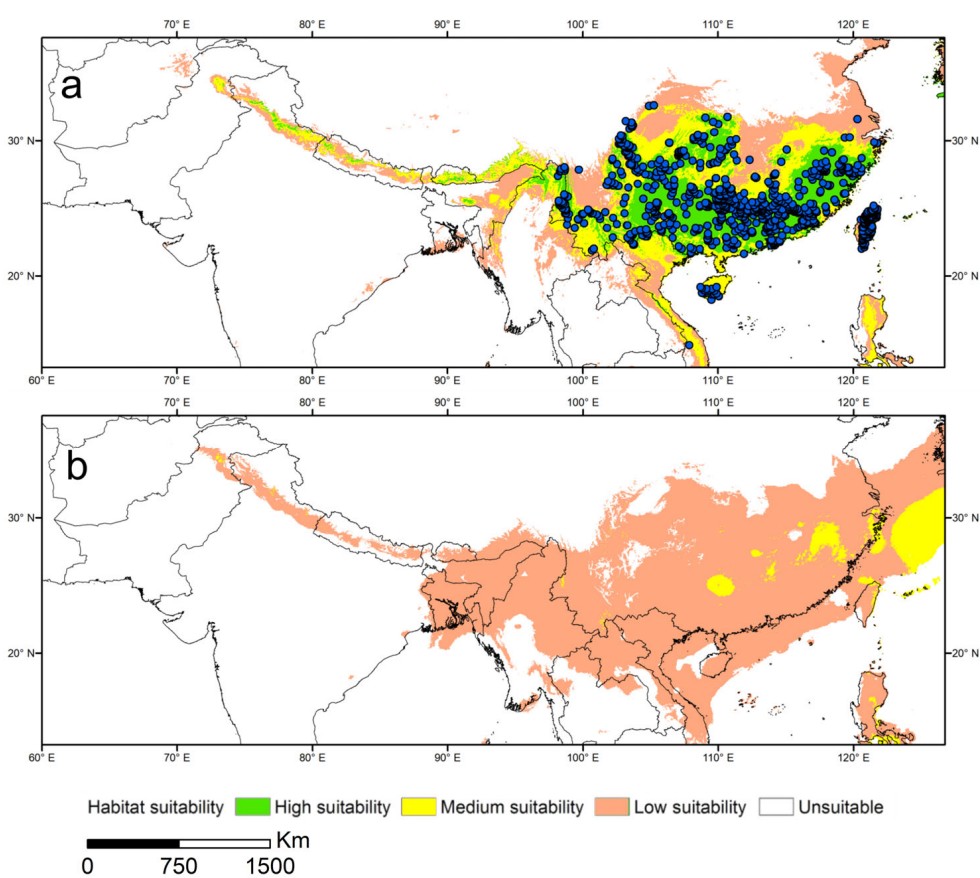

**Figure 3.** The simulated potential distribution areas of *Lithocarpus hancei* using RF model: (**a**) At the current stage; (**b**) At the LGM period. The blue dots show the occurrence records. The green areas indicate high-suitability areas; the yellow areas indicate medium-suitability areas; the pink areas indicate low-suitability areas, and the white areas indicate unsuitable areas.

### 3.2. Environmental Variable Contribution and Response Curves

Among the environmental variables, the precipitation of the warmest quarter showed the greatest contribution to the potential suitable distribution area of *L. hancei*, explaining 11.2% of its distribution. The precipitation of the driest month, the mean diurnal range, and temperature seasonality were also important to its distribution, accounting for 10.9%, 8.1%, and 7.6% of the variation, respectively (Table 1). Therefore, the main temperature variables affecting the current potential suitable area of *L. hancei* were MDR and TS, and the main

precipitation variables were PrecWQ and PrecDM. The contribution of soil and topographic variables was much lower than that of the other variables (Table S2), indicating that soil and topography only had a small overall impact on the current potential suitable area.

**Table 1.** The contribution of climatic variables of *Lithocarpus hancei* predicted by random forest model.

| Abbreviation | Description | Relative Variable Importance (%) |
|---|---|---|
| PrecWQ | Precipitation of Warmest Quarter/mm | 11.2 |
| PrecDM | Precipitation of Driest Month/mm | 10.9 |
| MDR | Mean Diurnal Range (Mean of monthly (max temp—min temp))/°C | 8.1 |
| TS | Temperature Seasonality (Standard Deviation × 100)/% | 7.6 |
| MTWM | Max Temperature of Warmest Month/°C | 3.5 |
| PS | Precipitation Seasonality (Coefficient of Variation) | 3.4 |
| AMT | Annual Mean Temperature/°C | 3.0 |
| PrecWM | Precipitation of Wettest Month/mm | 2.5 |

The main environmental variables that impacted the potentially suitable areas of *L. hancei* are illustrated in Figure S1. All of these variables showed a peak, indicating that *L. hancei* prefers habitats with the precipitation of the warmest quarter greater than 472.9 mm (with the optimum at 584.18 mm), the precipitation of the driest month greater than 13.08 mm (with the optimum at 48.59 mm), a mean diurnal range from 4.6 °C to 16.7 °C (with the optimum at 7.0 °C), and temperature seasonality from 96.7% to 739.7% (with the optimum at 412.93%).

### 3.3. Current Potential Distribution of Suitable Areas for L. hancei

The potential suitable area for *L. hancei* is mainly in East Asia up to 13°37′ N in the Northern Hemisphere, with a total area of about $315.89 \times 10^4$ km$^2$.

The spatial extent of the area of high suitability was $109.60 \times 10^4$ km$^2$ (Table 2), mainly located in South and East China, including Guizhou, Guangxi, Guangdong, Fujian, Taiwan, Southern Zhejiang, Southern Hunan, and Southern Jiangxi provinces, extending into Southwest China, the Southern Himalayas, and Northern Indo-China (Figure 3a). The spatial extent of the medium-suitability area was $98.02 \times 10^4$ km$^2$ (Table 2), mainly located in Southern China (Figure 3a). The spatial extent of the low-suitability area was $108.27 \times 10^4$ km$^2$ (Table 2), mainly located along the drainage basin of the Yangtze and Mekong Rivers, Vietnam, and the southern slope of the Himalayas (Figure 3a). The spatial extent of the area of unsuitability was $817.03 \times 10^4$ km$^2$ (Table 2). The detailed results for the present potential distribution of *L. hancei* are illustrated in Figure 3a.

**Table 2.** The suitable areas of *Lithocarpus hancei* at different time scales and under different future climate scenarios. (Units $10^4$ Km$^2$).

| Climate Scenario | | Unsuitability | Low Suitability | Medium Suitability | High Suitability | Suitable Area |
|---|---|---|---|---|---|---|
| LGM | | 860.39 | 395.91 | 31.84 | / | 427.76 |
| Current | | 817.03 | 108.27 | 98.02 | 109.6 | 315.89 |
| 2041–2060 | SSP1-2.6 | 797.84 | 195.33 | 140.5 | / | 335.83 |
| | SSP2-4.5 | 800.05 | 190.58 | 143.05 | / | 333.63 |
| | SSP3-7.0 | 796.17 | 209.82 | 127.68 | / | 337.51 |
| | SSP5-8.5 | 790.26 | 210.01 | 133.4 | / | 343.41 |
| 2061–2080 | SSP1-2.6 | 795.95 | 194.74 | 142.98 | / | 337.72 |
| | SSP2-4.5 | 787.44 | 206.53 | 139.7 | / | 346.23 |
| | SSP3-7.0 | 792.11 | 224.64 | 116.92 | / | 341.57 |
| | SSP5-8.5 | 776.65 | 230.36 | 126.66 | / | 357.02 |
| 2081–2100 | SSP1-2.6 | 783.83 | 206.86 | 142.98 | / | 349.84 |
| | SSP2-4.5 | 777.65 | 203.63 | 152.39 | / | 356.02 |
| | SSP3-7.0 | 763.11 | 234.07 | 136.49 | / | 370.56 |
| | SSP5-8.5 | 758.94 | 261.71 | 113.02 | / | 374.73 |

### 3.4. Potential Distribution of Suitable Areas for L. hancei during the LGM

Our simulation showed that the highly suitable area of *L. hancei* in the LGM period was reduced relative to the present. Only medium-suitability habitats were located in Southeast China, with a spatial extent of $31.84 \times 10^4$ km$^2$ (Table 2). The low-suitability habitats were situated along the continental shelf of the East China Sea to mainland China and Central China, with a spatial extent of $395.91 \times 10^4$ km$^2$ (Table 2). The spatial extent of the unsuitable area was $860.39 \times 10^4$ km$^2$ (Table 2, Figure 3b). The detailed potential distribution map for *L. hancei* during the LGM is shown in Figure 3b.

### 3.5. Potential Distribution of Suitable Areas for L. hancei in the Future

The predicted distribution of *L. hancei* in response to four SSPs during three future periods (2041–2060, 2061–2080, and 2081–2100) is shown in Figure 4. Similar to the distribution during the LGM period, the future highly suitable habitable environments will be lost under all the climate scenarios in the future. The medium suitable area under the SSP2-4.5 scenario is predicted to decrease first and then increase, and the opposite is true for SSP5-8.5. Under the SSP3-7.0 scenario, it is predicted to increase, then decrease, and ultimately increase again, but it might experience a continuous decrease under the SSP1-2.6 scenario (Figure 4, Table 2). In the 2041–2060 period, the medium-suitability area of the SSP2-4.5 scenario is the largest, followed by those of the SSP1-2.6, SSP 5-8.5, and SSP3-7.0 scenarios. In the 2061–2080 period, the SSP1-2.6 scenario medium-suitability area is predicted to be the largest, followed by those of the SSP2-4.5, SSP5-8.5, and SSP3-7.0 scenarios. In the 2081–2100 period, the SSP2-4.5 scenario medium-suitability area might be the largest, followed by those of the SSP1-2.6, SSP3-7.0, and SSP5-8.5 scenarios. The predicted medium suitable area of 2081–2100 in the SSP5-8.5 scenario was the smallest. The low-suitability areas under the SSP1-2.6 scenario are predicted to increase first and then slightly decrease, followed by a final decrease. Under the SSP2-4.5 scenario, the overall suitable area will increase first and then decrease, while the low-suitability areas under the SSP2-4.5, SSP3-7.0, and SSP5-8.5 scenarios are predicted to continue to increase (Figure 4, Table 2). The unsuitable areas will be reduced in all future scenarios (Figure 4, Table 2).

In general, the highly suitable area of *L. hancei* is expected to be lost in all future scenarios. However, the medium-suitability area might be increased dramatically, compared to the present distribution, with a trend of northward movement. The low-suitability and unsuitable areas are expected to shift to the north.

### 3.6. Spatial Delineation of Range Contraction or Expansion

We obtained relative changes in the suitable area for *L. hancei* by comparing the potential distribution areas between the current and past climate scenarios (Figure 5) and between the future and current climate scenarios (Figure 6).

Our prediction showed that during the LGM period to the present, the contraction area of *L. hancei* was much larger than the expansion area. Therefore, the species experienced an overall range expansion (Table 3). The newly gained areas during this episode were mainly located in marginal areas of the current distribution (central and east China, the Southern Himalayas, and Northern Indo-China) (Figure 5).

Comparing the present and the future simulation results revealed the distribution dynamic showed an overall expansion trend, as the area of contraction was smaller than the area of expansion (Table 3). The expansion areas were mainly in Southwest China and East China (including Sichuan, Tibet, the junction of Jiangsu, Hubei, and Henan, and Northern Anhui). The lost areas were mainly within Central China, the Southern Himalayas, and Northern Indo-China (Figure 6).

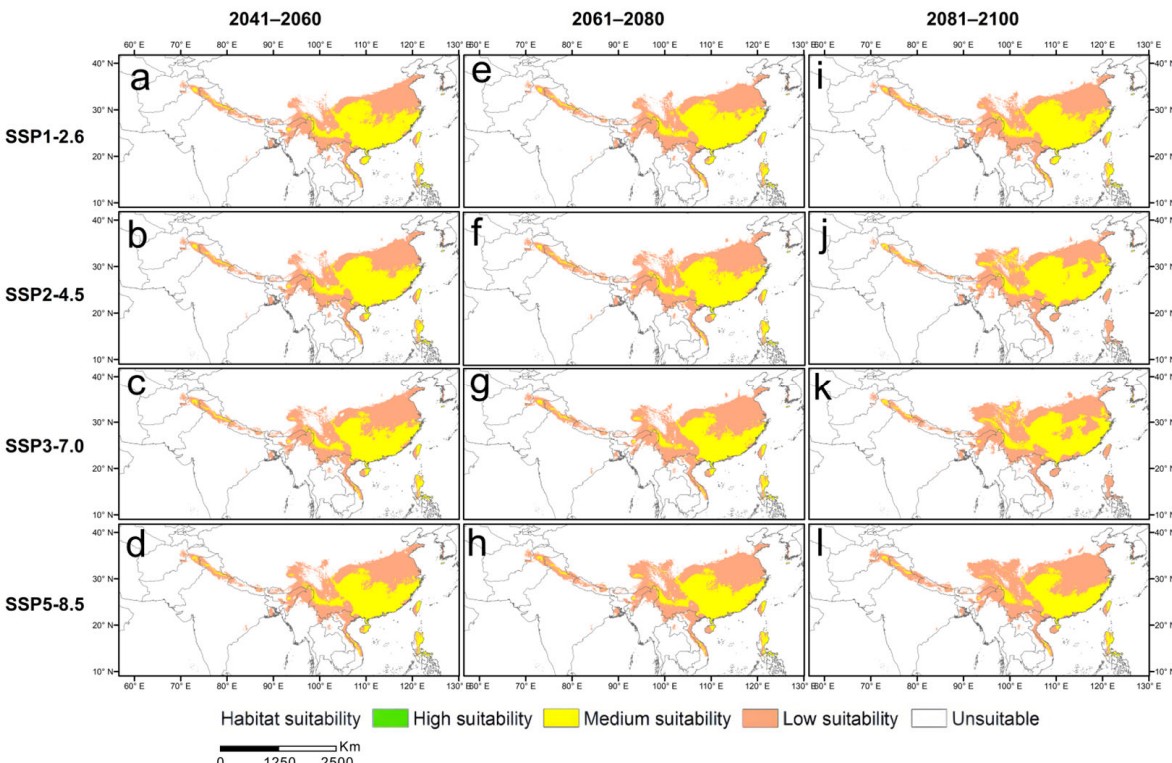

**Figure 4.** Potential distribution area of *Lithocarpus hancei* under different periods and climate scenarios in the future. (**a**–**d**) The potential distribution area of *L. hancei* at 2041–2060 under the four shared socioeconomic pathways SSP1-2.6 (**a**), SSP2-4.5 (**b**), SSP3-7.0 (**c**), SSP5-8.5 (**d**), respectively; (**e**–**h**) The potential distribution area of *L. hancei* at 2061–2080 under four shared socioeconomic pathways SSP1-2.6 (**e**), SSP2-4.5 (**f**), SSP3-7.0 (**g**) and SSP5-8.5 (**h**), respectively; (**i**–**l**) The potential distribution area of *L. hancei* at 2081–2100 under four shared socioeconomic pathways SSP1-2.6 (**i**), SSP2-4.5 (**j**), SSP3-7.0 (**k**) and SSP5-8.5 (**l**), respectively.

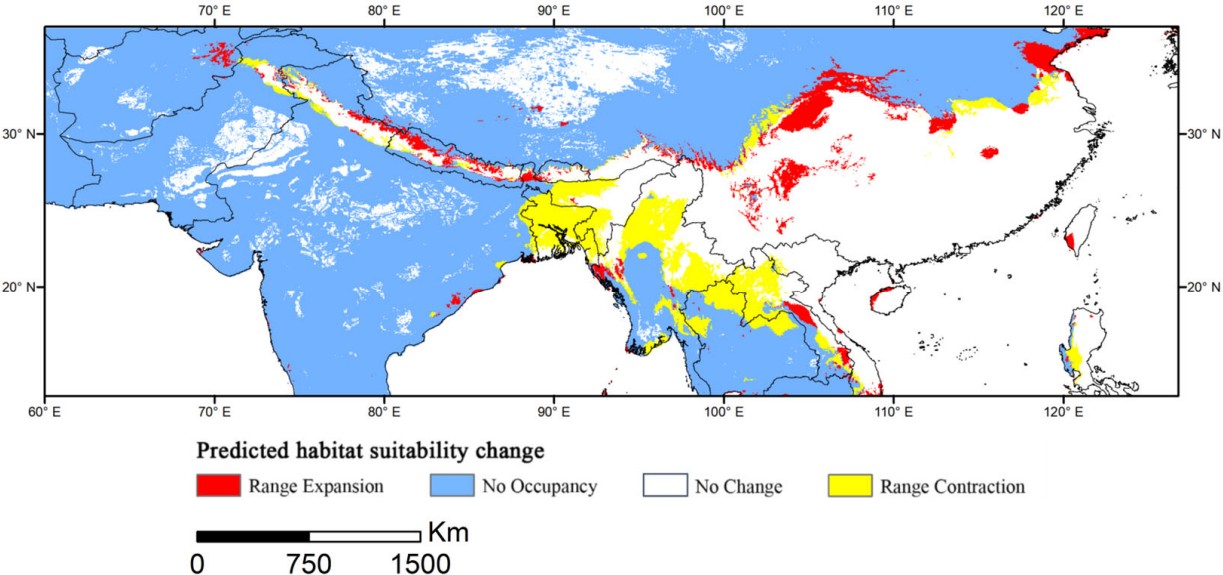

**Figure 5.** Changes in the suitable area of *Lithocarpus hancei* at the LGM. The red areas indicate range expansion; the blue areas indicate no occupancy; the white areas indicate no change, and the white areas indicate range contraction.

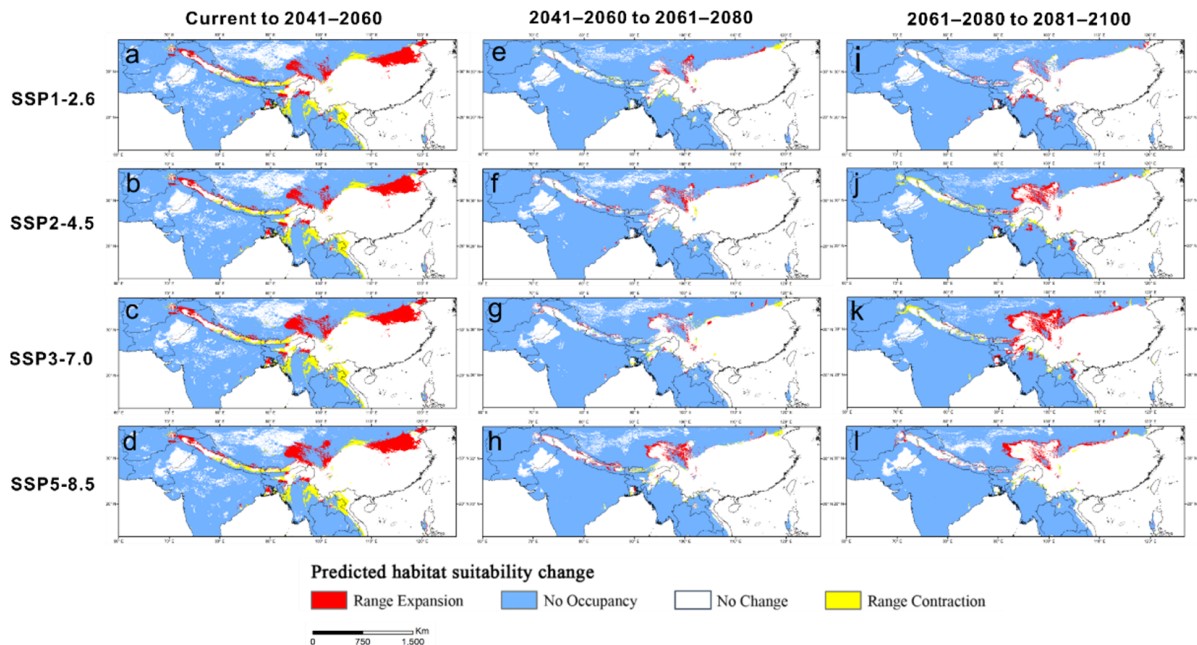

**Figure 6.** Changes in the suitable area of *Lithocarpus hancei* at three time slices under four $CO_2$ emission scenarios in the future. (**a–d**) The changes in the suitable area of *L. hancei* form current to 2041–2060 under the four shared socioeconomic pathways SSP1-2.6 (**a**), SSP2-4.5 (**b**), SSP3-7.0 (**c**), SSP5-8.5 (**d**), respectively; (**e–h**) the change in the suitable area of *L. hancei* from 2041–2060 to 2061–2080 under four shared socioeconomic pathways SSP1-2.6 (**e**), SSP2-4.5 (**f**), SSP3-7.0 (**g**) and SSP5-8.5 (**h**), respectively; (**i–l**) the change in the suitable area of *L. hancei* from 2061–2080 to 2081–2100 under four shared socioeconomic pathways SSP1-2.6 (**i**), SSP2-4.5 (**j**), SSP3-7.0 (**k**) and SSP5-8.5 (**l**), respectively.

**Table 3.** Distribution changes of *Lithocarpus hancei* based on the binary distribution under different climate scenarios in comparison to the current binary distribution. (Units $10^4$ $Km^2$).

| Climate Scenario | Range Expansion | No Occupancy | No Change | Range Contraction |
|---|---|---|---|---|
| LGM to current | 43.66 | 716.23 | 304.46 | 71.52 |
| Current to 2041–2060 SSP1-2.6 | 60.24 | 707.61 | 303.88 | 39.47 |
| Current to 2041–2060 SSP2-4.5 | 59.7 | 707.48 | 301.66 | 41.69 |
| Current to 2041–2060 SSP3-7.0 | 66.5 | 699.71 | 298.76 | 44.55 |
| Current to 2041–2060 SSP5-8.5 | 69.24 | 697.42 | 302.17 | 41.18 |
| 2041–2060 to 2061–2080 SSP1-2.6 | 10.67 | 805.41 | 357.88 | 8.94 |
| 2041–2060 to 2061–2080 SSP2-4.5 | 17.13 | 791.34 | 360.34 | 3.75 |
| 2041–2060 to 2061–2080 SSP3-7.0 | 12.05 | 792.84 | 360.25 | 7.74 |
| 2041–2060 to 2061–2080 SSP5-8.5 | 22.72 | 778.16 | 365.9 | 8.35 |
| 2061–2080 to 2081–2100 SSP1-2.6 | 15.39 | 799.33 | 366.5 | 2.05 |
| 2061–2080 to 2081–2100 SSP2-4.5 | 27.23 | 785.5 | 360.64 | 16.83 |
| 2061–2080 to 2081–2100 SSP3-7.0 | 42.04 | 777.82 | 361.04 | 11.26 |
| 2061–2080 to 2081–2100 SSP5-8.5 | 24.99 | 766.83 | 382.07 | 6.55 |

*3.7. Shifts in the Distribution Centroid*

The shifts of the centroid of *L. hancei* during different periods under different $CO_2$ emission scenarios are illustrated in Figure 7. Overall, the centroids of the suitable habitats will stretch to the north in all future scenarios, but show some difference in distance and direction under different $CO_2$ scenarios.

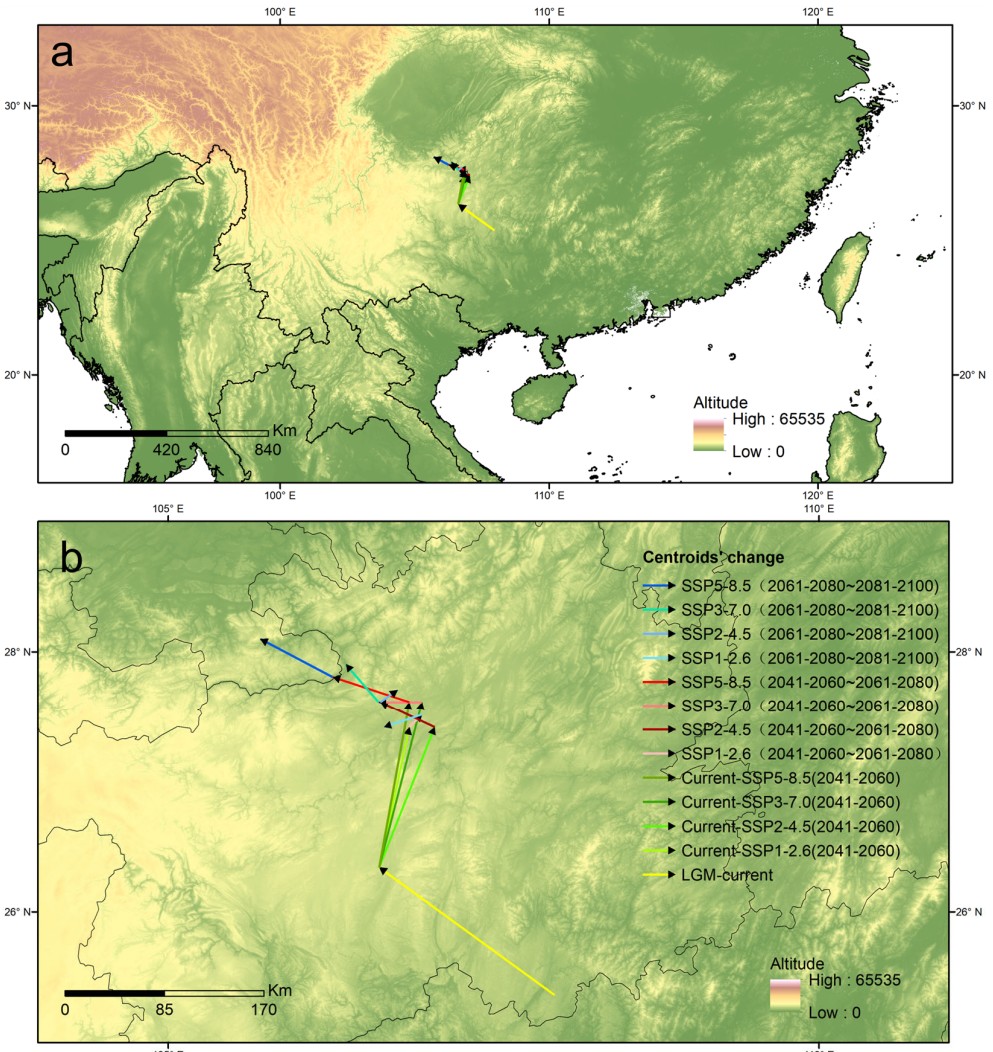

**Figure 7.** The predicted distribution centroids shift of *Lithocarpus hancei* at the LGM to current, the current to 2041–2060, 2041–2060 to 2061–2080, and 2061–2080 to 2081–2100: (**a**) General distribution locations of centroid; (**b**) accurate centroid shifts of suitable areas. The centroid shift direction and distance at different periods and specific scenarios were illustrated by the colorful lines and arrows listed at the right side of the image.

## 4. Discussion

### 4.1. The Main Environmental Variables Restricting the Distribution of L. hancei

Precipitation and temperature are the most important bioclimatic factors affecting tree species distributions [34]. The predicted high-suitability areas of *L. hancei* are located in East Asia, especially in East China and South China. There are few distribution areas in Northern Myanmar and the southern slope of the Himalayas. These results are consistent with the actual distribution of *L. hancei*, indicating our simulation is successful and accurate.

Our prediction results showed that the precipitation of the warmest quarter, the precipitation of the driest month, the mean diurnal range, and temperature seasonality were the dominant factors affecting the distribution of *L. hancei*. Among them, the precipitation of the warmest quarter contributed the most to explaining the species distribution. This result was similar to other widespread fagaceous trees in the East Asia subtropics. For example, compared with the LGM, the present distributions of the evergreen oak *Quercus glauca* (section *Cyclobalanopsis*) in Southwest China and the lowlands on the southern slopes of the Himalayas showed a slight contraction; precipitation during the warmest quarter accounted for more than half of the variation (account 62.27%) in its distribution [35]. A similar

scenario was also found for another East Asian widespread deciduous oak, *Q. acutissima* (section *Cerris*), such that the precipitation of the warmest quarter was the most important environmental variable impacting its distribution [36]. Precipitation directly provides plants' water supply, especially in summer, when transpiration demand is high. A lack of water can lead to embolization in the vasculature and thus affect plant growth, development, and other processes, even leading to plant death [37]. Likewise, the precipitation of the driest month also affected the survival of *L. hancei*. Our field survey showed that *L. hancei* could grow in areas with a prominent dry season, where the precipitation in the driest month is only 13.08 mm. Therefore, *L. hancei* can tolerate intermediate drought stress. This result also agrees with our field observation that *L. hancei* has a dwarf shrubby form on the sunny slope of limestone areas, where seasonal or temperate drought is prominent. In contrast, the species can grow better in an environment with deep soil layers composed of fertile and loose soil and in moist woodlands [38].

The northern distribution limits of the woody species were restricted by temperature factors [39]. Our study found that the mean diurnal range and temperature seasonality also impacted the distribution of *L. hancei*. The average diurnal temperature difference optimal for the *L. hancei* distribution was 7 °C. Our simulation predicted that there are consistently fewer suitable distribution areas in the northern region across all the periods concerned, which may be owing to the significant temperature difference between day and night in the north portion of the study area. Similarly, the northern distribution limits of other fagaceous trees, e.g., *Quercus fabri* [40], *Q. variabilis* [41], *Q. chungii* [42], *Castanopsis fissa* [43], and *C. carlesii* [44], were also restricted by temperature, but deciduous trees have better cold tolerance (with an optimal mean diurnal temperature of 6.5–9.0 °C) [45] than evergreen subtropical trees (with an optimal mean diurnal temperature of 10.9–13.5 °C) [46].

*4.2. Suitable Habitats for L. hancei under the Current Climate*

Under the current climate, *L. hancei* has a broad regional distribution, mainly within East Asia. The highly suitable habitats are concentrated primarily in East and South China, regimes by the East Asian Summer Monsoon (EASM) region, characterized by a warm climate with a hot summer, constant humidity, and abundant precipitation. The total precipitation decreases from the southeast coast to the northwest inland. The annual average temperature is 15–22 °C, and the average temperature of the coldest month is 0–15 °C [47]. A warm and humid climate is crucial for the healthy growth of *L. hancei*. The climatic conditions in East and Southern China well match the growth criteria of *L. hancei*. The medium- and low-suitability areas of *L. hancei* have a warm climate with a hot summer (the average temperature of the hottest month is greater than 22 °C), but the precipitation is concentrated in summer and the winter is very dry, which is not the best habitat for *L. hancei*. In addition, Southwest China, the southern slope of the Himalayas, and the Indo-China regions are under an Indian summer monsoon regime [48], where the temperature difference between day and night is more significant, and the seasonal drought (September–April) is more prominent than the EASM [49]. Likewise, a long seasonal drought during the year is unsuitable for the long-term survival of *L. hancei*. Thus, suitable areas for *L. hancei* in the west are rare.

*4.3. Changes in the Suitable Habitats of L. hancei under Future Climate Scenarios*

The high-suitability areas of *L. hancei* experienced a prominent expansion from the LGM to the present, with contractions of the medium- and low-suitability habitats (Figure 3). The climate extremes indeed had a significant impact on *L. hancei*. During the LGM, the global climate experienced drastic changes [50]. Although the continental ice sheet was not developed in East Asia, drought, semiarid conditions, and reduced sea levels exposing continental shelves resulted in range contractions and a southern retreat of thermophilic biota [51]. Our results showed that the suitable temperature range of *L. hancei* is 8.8–21.5 °C (Figure S1). The low temperature during the LGM might have been the key factor restricting the distribution of *L. hancei*.

The future potential suitable areas for *L. hancei* are mainly in Southeast China, and the highly suitable areas will dramatically be lost or severely degraded. Climate models predict that Earth's global average temperature will rise at least 1.5 to 2 °C by the end of the century [52]. Additionally, the conventional drought assessments of soil moisture projections show that Southeastern China will become a drought hotspot in East Asia in the late decades of the 21st century [53]. The increased temperature and decreased precipitation in the core distribution area will cause habitat degradation due to significant drought stress [54,55]. While higher $CO_2$ concentrations may boost plant growth and help plants conserve water [56], the co-occurrence of high temperatures during drought could exacerbate plant stress [57]. The decrease in precipitation and an increase in temperature in the southeastern distribution areas of *L. hancei* in the future might be key drivers of such demographic changes.

Moreover, the centroid of *L. hancei* will shift to the north in the future. Other evergreen oaks of *Quercus* section *Cyclobalanopsis* also show a northward expansion tendency [58]. Notably, the predicted total suitable habitats of *L. hancei* tend to expand towards higher elevations in its western distribution range. Increased precipitation in these regions may improve high-altitude suitability [59]. Likewise, simulation of the distribution of *Taxus wallichiana* shows that rising temperatures and extreme precipitation events may induce disruptions in its physiology. For example, under the RCP8.5 scenario, there is predicted to be a steady decline in the suitable climatic habitat of *T. wallichiana*. The changes are more apparent in the Himalayas and Southeast China region. These results also predicted that the distribution of *T. wallichiana* may shift to higher elevations under all climate scenarios [60]. Although the distribution dynamics differed among the four shared socioeconomic pathways, our results revealed a general trend of a massive shift of the highly suitable habitats to medium- and low-suitability areas of *L. hancei* (Table 2). Overall, the habitat of *L. hancei* will experience significant degradation in the future.

MMEBFs often develop on mountain tops or ridge lines [61]. Hence, the high heterogeneity of mountainous areas can buffer the impact of extreme climate conditions to allow species within ecotones to shift along an altitude to migrate into suitable habitats [62]. Elevational range shifts are currently the most commonly documented response to historical climate change [63]. A similar situation has been identified in our previous work on *Quercus arbutifolia*, an endangered oak tree endemic to MMEBFs, and among its five known populations, only one in Fujian is predicted to migrate into a highly suitable area in the future; thus, the habitats of the remaining populations might completely disappear or will shrink dramatically [64]. As temperatures continue to rise, *Lithocarpus* species are more likely to occur at higher elevations. The suitable habitats of MMEBFs are doomed to shrink or be degraded, thus pushing this unique ecosystem to a more vulnerable status.

East Asian MMEBFs trees show similar demographic changes, distribution dynamics, and responses to climate scenarios. For example, under a warming climate scenario, the distribution area of *Tsoongiodendron odorum* (Magnoliaceae) is projected to decrease, and especially the highly suitable area is predicted to be significantly reduced [65]. Likewise, compared with the current climate scenarios, climate change in the future may reduce the high suitability habitat for tea (*Camellia sinensis*) (Theaceae) and increase the area of medium- and low-suitability habitats [66]. These different studies all suggest MMEBFs will be degraded under all scenarios in the future, which makes MMEBFs one of the most vulnerable forestry ecosystems in the Asian subtropics. Therefore, the conservation of MMEBFs is urgent and should be enforced.

### 4.4. Conservation and Forestry Management of East Asia MMEBFs

As a long-lived tree species with moderate tolerance to seasonal drought, *L. hancei* can cope with the future stressful climatic conditions in the short term in their current populations, which could result in a prominent time lag before the signs of habitat degradation are obvious among the populations. It should be noted that fagaceous seeds are generally recalcitrant and lose their viability even under mild drought conditions [67–69].

Moreover, they are also typically not suitable for long-distance dispersal [70]. As dry season precipitation continues to decrease, populations of *L. hancei* will face pronounced difficulty in seedling establishment and regeneration. As MMEBFs will gradually degrade in the future, the manual transplantation of seedlings back into populations as well as the introduction of germplasm into the future suitable areas in the north and northeast of the study area should be conducted.

The distribution of *L. hancei* in Southwest China, Indo-China, and the Southern Himalayas is scattered and highly fragmented. Recent phytogeographic research on other subtropic evergreen fagaceous trees, e.g., *Quercus glauca* [35], *Q. franchetii* [71], and *Q. schottkyana* [72,73], has revealed very different local adaptation abilities, limited gene flow, and unique genetic compositions of the marginal populations compared to that in the core distribution area. Given their limited distribution and small size, these marginal populations will become more vulnerable under global climate change scenarios. Therefore, ex situ conservation, introducing unique germplasm of marginal populations into adjacent suitable habitats, and developing effective cryopreservation protocols are essential to preserve the genetic diversity and long-term viability of *L. hancei* populations.

Notably, our prediction reveals the general distribution dynamics of *L. hancei* under future climate change scenarios to inform forestry conservation. Other factors, e.g., human disturbance, wildfire, etc., also profoundly impact these forest trees. Future research coupling SDMs and 3S-based technology to monitor land use and forest health can provide more accurate and crucial information on the conservation of the MMEBFs in East Asia.

## 5. Conclusions

East Asia MMEBFs are biodiversity and conservation hotspots, but their unique micro-environmental envelope and highly fragmented habitats make this ecosystem highly vulnerable. Our study on the widespread Asian MMEBFs tree species *L. hancei* showed that the precipitation of the warmest season, temperature seasonality, the monthly average temperature difference between day and night, and precipitation in the driest month have the most significant contribution to its distribution. The highly suitable habitats for the species are concentrated in South and East China, which indicates that *L. hancei* prefers warm and humid climates. In future climatic scenarios, its high-suitability areas in the southeast and the west will decrease with centroids shifting to the north. *L. hancei* is a long-lived tree with moderate tolerance to mild to medium seasonal droughts once a seedling has been established. Although populations in low-suitability areas may survive for a short time, the sensitivity of acorns/seeds to drought may hinder population regeneration, resulting in ecosystem degradation. Considering the limited dispersal ability of its acorns and the fragmentation and isolation of their habitats and populations, the threatened status of Asian MMEBFs tree species is much higher than previously assessed. In situ conservation in the long-time stable habitats of *L. hancei* and enlarging the protected region to cover its future core distribution areas, plus ex situ conservation to establish germplasm collection and transplantation of seedlings to their future high-suitability habitats in the north, provide efficient solutions to secure the long-term survival of *L. hancei* and the unique Asian MMEBFs.

**Supplementary Materials:** The following supporting information can be downloaded at: https://www.mdpi.com/article/10.3390/f14051049/s1, Figure S1: Response curves of the eight selected climatic variables that were used in SDMs for Lithocarpus hancei, Table S1: Records of Lithocarpus hancei were obtained after thinning, Table S2: Percent contributions of the bioclimatic and topographic variables in the species distribution models for Lithocarpus hancei, Table S3: Results of Pearson correlation analysis of 19 bioclimatic variables for Lithocarpus hancei, Table S4: Evaluation results of species distribution models for *Lithocarpus hancei*.

**Author Contributions:** Conceptualization: Y.Y., L.L., Y.T. and M.D.; data curation: Y.Y.; formal analysis: Y.Y.; methodology: Y.Y., L.L., Y.T. and M.D.; software: Y.Y. and L.L.; visualization: Y.Y.; writing—original draft: Y.Y.; writing—review and editing: Y.Y., L.L., Y.T. and M.D. All authors have read and agreed to the published version of the manuscript.

**Funding:** This research was funded by "the National Scientific Foundation of China (NSFC), grant number 31972858, granted to Min Deng and grant number 31970223, granted to Yunhong Tan", "Yunnan Key Laboratory for Integrative Conservation of Plant Species with Extremely Small Populations, grant number PSESP2021, granted to Min Deng", "the Southeast Asia Biodiversity Research Institute, Chinese Academy of Sciences, grant number Y4ZK111B01, granted to Yunhong Tan", "the project of the Yunnan Academy of Forestry and Grassland, grant number KFJJ21-05, granted to Min Deng".

**Data Availability Statement:** Not applicable.

**Acknowledgments:** We are thankful to Asian Elephant Yunnan Field Scientific Observation and Research Station, Yunnan Asian Elephant Field Scientific Observation and Research Station of the Ministry of Education, and Baima Snow Mountain Complex Ecosystem Vertical Transect Field Observation and Research Station for their help on the field work.

**Conflicts of Interest:** The authors declare no conflict of interest.

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
