# Peer review of "How Climate Change Impacts the Distribution of Lithocarpus hancei (Fagaceae), a Dominant Tree in East Asian Montane Cloud Forests"

_forests, doi:10.3390/f14051049_

Round 1

Reviewer 1 Report

Comments for Author,

I have read your paper carefully. This paper aimed to How Climate Change Impacts the Distribution of Lithocarpus hancei (Fagaceae), a Dominant Tree in East Asian Montane Cloud Forests. Even the paper is interesting and scope of Forests journal. I have some minor comments.

Specific comments:

1- Abstract could be more comprehensive. So, Please give more information regarding Impacts the Distribution

2- I think, there are many unnessarry references in the introduction and discussion. Please use more properly references.

3- Discussion is superficial. Please give more attention for your obtained results.

4- Even Author mentioned regarding climate change detrimental effects in the title, introduction. I could not see any real discussion detrimental effects of climate change and which factors induce the climate change sources. Please support this section dramatically. 

6- Conclusion remark seems generally. Please specify this section and add your recommandation.

Best Regards

Please recheck all paper. Because, there are many gramaticall errors thoroughly paper. 

Reviewer 2 Report

Impact of Climate change on land, forest and land use and management is at present time solved on a large scale. I appreciate the choice of topic, chosen methods of solution, appropriate presentation, exhaustive discussion and well-elaborated conclusion.

Presented paper is by my opinion good prepared and designed in standard requirements for Forests Journal. I´m satisfied with overall Quality of Article. Abstract is comprehensive, the Introduction Chapter is adequate, broad and clear. Used Materials and Methods with proposed Research Design are suitable and adequate. Results fulfil scientific standard and Discussion is too adequate for this Article. Authors have used by above standard number of references.

I have some next small comments and recommendations, which by my Opinion target to improve to final quality of this Article:

 1. For me is little unstandardized mark of two correspondence authors, I think, only one from those register Manuscript, I recommend mark only one Correspondence Author. Btw, why were used by Yunhon Tan Author Initials Y. H. instead Y. T.?

 2. Authors have used abbreviation for Evergreen broadleaved forests (EBLFs), I think that sufficient is EBFs (corresponding with MMEBF). I recommend use this in each occurrence in the text.

 3. Figure 2 presents by my Opinion Results (modelled suitability), I think in Methods is sufficient Map with current distribution of L. hancei. I recommend move Map models to Fig. 3.

 4. I have comments by description of Variable selection (2.3 and Table 1). For me is used Description (bio1-bio18) lightly un-transparent or redundant, instead “bio” I recommend use abbreviations of selected Variables, e.g. Precipitation of Warmest Quarter – PrecWQ, or easy only serial number (1-8.).

 5. I do not fully understand term “temperature seasonality with values 412 or 739% (row 252), are those value (percentwise) all right correct?

 6. For easer and better understanding I recommend by Fig. 3 and Fig. 5 append in the top appertaining year (so top left 2041-2060, top centre 2061-2080, top right 2081-2100).

 7. Chapter 3.7 is for me little informatics redundant (to importance of whole paper with 6 text rows), I recommend present only 1 map, or adequately reduce size.

Reviewer 3 Report

This study is very interesting and highlights the possible response to future climate change in mid-latitude broadleaf forest ecosystems.

The topic is also important for the management of strategic resources for the future of the planet.

The methodology based on the correlation of multidisciplinary variables is correct and has produced interesting and original results.

I suggest improving the definition of the images by decreasing the scale of representation of the maps to see more important details for verifying the results.

Other tips / reviews:

line 315: there is a repetition of "white" that needs to be replaced with "yellow"

minor editing of English language required

Round 2

Reviewer 1 Report

Author improved paper carefully, now the paper could be accepted for the publication.

Best Regards